# Preparation and Size Control of Efficient and Safe Nanopesticides by Anodic Aluminum Oxide Templates-Assisted Method

**DOI:** 10.3390/ijms22158348

**Published:** 2021-08-03

**Authors:** Chunxin Wang, Bo Cui, Yan Wang, Mengjie Wang, Zhanghua Zeng, Fei Gao, Changjiao Sun, Liang Guo, Xiang Zhao, Haixin Cui

**Affiliations:** Institute of Environment and Sustainable Development in Agriculture, Chinese Academy of Agricultural Sciences, Beijing 100081, China; wangchunxin@caas.cn (C.W.); cuibo@caas.cn (B.C.); wangyan03@caas.cn (Y.W.); wangmengjie@caas.cn (M.W.); zengzhanghua@caas.cn (Z.Z.); gaofei@caas.cn (F.G.); sunchangjiao@caas.cn (C.S.); guoliang01@caas.cn (L.G.)

**Keywords:** buprofezin, nanoparticles, AAO template, tunable particle size

## Abstract

Efficient and safe nanopesticides play an important role in pest control due to enhancing target efficiency and reducing undesirable side effects, which has become a hot spot in pesticide formulation research. However, the preparation methods of nanopesticides are facing critical challenges including low productivity, uneven particle size and batch differences. Here, we successfully developed a novel, versatile and tunable strategy for preparing buprofezin nanoparticles with tunable size via anodic aluminum oxide (AAO) template-assisted method, which exhibited better reproducibility and homogeneity comparing with the traditional method. The storage stability of nanoparticles at different temperatures was evaluated, and the release properties were also determined to evaluate the performance of nanoparticles. Moreover, the present method is further demonstrated to be easily applicable for insoluble drugs and be extended for the study of the physicochemical properties of drug particles with different sizes.

## 1. Introduction

Pesticides are widely used in protecting crops and improving crop yield to meet agricultural production [1,2]. However, the low effective utilization rate of pesticides is a serious problem. Therefore, improving the utilization rate of pesticides has become the focus of pesticide formulation research [3]. According to Noyes–Whitney equation, the dissolution rate of drugs is directly proportional to the surface area of drug particles, and therefore the most effective method is to reduce the particle size of drug particles [4]. Nanopesticide formulations have been developed to maximize pesticide utilization while minimizing the side effects [5,6,7,8].

Currently, the research of the nanopesticide falls into two categories: one is to use nanotechnology including emulsification and homogenization to prepare nanoparticles and the other is to use nanomaterials to construct nanoparticles [9,10,11]. The former is mainly affected by external factors including equipment and process parameters, it faces critical challenges including relatively large particle size difference, batch differences and low production efficiency [12,13,14,15]. Functionalized abamectin poly (lactic acid) nanoparticles were prepared by the ultrasonic emulsification to explore the adhesion of nanoparticles to the leaf surface of crops. Antagonistic effect of azoxystrobin poly (lactic acid) microspheres with controllable particle size was studied. These studies showed a large influence of the preparation conditions on the drug particles [16,17]. The latter can accurately control the particle size and improve the production efficiency [18,19,20,21,22]. Pyrimethanil-loaded mesoporous silica nanoparticles was synthesis to explore the distribution and dissipation in cucumber plants. Abamectin using porous silica nanoparticles as carriers were constructed to evaluate controlled-release performance. The nanoparticles prepared by nanomaterials can ensure the uniformity of particle size [23,24].

As known to us, anodized aluminum oxide (AAO) templates have been widely used for forming nanotubes and nanowires. The tiny holes on the surface of AAO are arranged in an orderly manner. In addition, the pore channels of the AAO template were parallel to each other, and the pore size uniformity of the template was good. Based on the above principle, the AAO template has been applied for forming nanoparticles by making full use of the characteristics of AAO template. However, AAO templates have never been used for preparing pesticide nanoparticles. In this work, we demonstrated for the first time that the AAO template method can in fact be extended for preparing uniform pesticide nanoparticles comparing to the conventional reprecipitation approach [25,26,27,28].

Buprofezin as an insect growth regulator has the characteristics of high activity, high selectivity and long residual period, which can inhibit chitin synthesis and interferes with metabolism in insect pests [29]. However, insecticidal application of poorly soluble drugs, such as buprofezin, in farmland systems is limited. In addition, it will directly pollute the environment after large-scale application of buprofezin [30]. Therefore, developing nanopesticide formulations and clarifying the effects of particle size on physicochemical properties and biological activity of pesticides is the key to solve the problem of poor dispersibility of insoluble pesticides [31,32,33,34,35].

In this study, the buprofezin nanoparticles (BNPs) with different sizes were developed via AAO template-assisted process. The particles with similar surface characteristics and extremely narrow size distribution were obtained using AAO templates with different pore sizes, which could also exclude the impact of shape and surface charge compared with the traditional method. The effect of drug concentration and number cycles on the size of particle was investigated. At the same time, the stability and release properties of nanoparticles were evaluated to characterize the effects of particle size and dispersion on their physicochemical properties. In conclusion, this research not only established an innovative and simple method for the potential application of nanopesticides with controllable particle size, excellent monodispersity and uniform morphology in plant protection, but also laid a foundation for the study of the physicochemical properties of drug particles with different sizes.

## 2. Results and Discussion

### 2.1. Characterization of the BNPs

In the BNPs fabrication process, BNPs were prepared using 100 nm AAO template in Figure 1. The mean particle size of BNPs based on a scanning electron microscope (SEM) image (Figure 1b) and a transmission electron microscope (TEM) image (Figure 1c) were 100 nm and 103 nm, respectively, which was smaller than the hydrated particle size (Figure 1a). SEM and TEM shows the real particle size in a dried state, whereas dynamic light scattering (DLS) provides the micelle core and swollen corona [36,37].

These results indicated that BNPs exhibited almost spherical morphology and comparatively monodisperse distribution. As known to us, the size of each aperture of the template is consistent, and the template is arranged in order, so the prepared drug particles are uniform in size. During the whole preparation process, drug molecules were gradually deposited in the pore of AAO template, and the particle size was limited by the pore size finally [25,27]. Therefore, the template with different pore size can be used to prepare nanoparticles with different sizes, which laid a foundation for the study of the physicochemical properties of drug particles with different sizes.

The SEM images of AAO templates after immersing in methanol, tetrahydrofuran and acetone were shown in supporting Information Appendix A. The pore size of the template immersed in tetrahydrofuran and acetone increased to 132 nm, which changed the structure of the template and affected the preparation of drug particles with different sizes. The main reason is that the structure of the template was destroyed in tetrahydrofuran and acetone solution, but this did not happen in methanol solution. The pore size of the template immersed in methanol has not changed. Therefore, methanol was selected as the solvent of the drug.

To confirm the difference between the properties of BNPs prepared with the present method and free drug molecules. We also developed buprofezin particles (BPs) using reprecipitation. The shape of drug particles was irregular and the size distribution of the particles was not uniform in Appendix A.

### 2.2. Optimization of Preparation Parameters

The optimization of the preparation process is very important for the size and distribution of the drug particles. As shown in Figure 2a, the AAO templates with pore size of 100 nm were immersed in the lower concentration of the drug solution, which can form smaller nanoparticles. The particle size of the BNPs with 20 mg/mL was larger than that of the BNPs with 5 and 10 mg/mL. The active components entering into the pore size of the template in unit time increased with the increase of drug concentration, thus enlarging the particle size. The particle size of the particles varied little at the concentration of 20 and 40 mg/mL. In a certain concentration range, the size of drug particles became larger with the increase of the concentration of solution, but it was always controlled in the range of template pore size. It was also observed that the particle size increased with the number of cycles (2, 5, 10 cycles). The particle size changed little at 5 and 10 cycles. The drug particles will grow along with the increase of the number of cycles, but the size of the particles is always affected by the pore size of the template. Overall, the amount of drug nanoparticles is less in the case of low concentration and low cycle times, which is not suitable for mass production. In conclusion, the optimal preparation conditions for BNPs were as follows: drug concentration (20 mg/mL), cycle number (5 cycle) and actuation duration (5 min).

### 2.3. Release Behavior

BNPs with 100 nm size were selected to confirm the better physicochemical properties comparing to the free drug molecules. The drug release profile is of importance in applying the proposed system for practical drug delivery. Figure 3 indicated that the prepared BNPs had characteristics of rapid release compared with free buprofezin particles prepared by the reprecipitation method, which was due to the dissolution rate. The cumulative release rate of the BPs was only 40.2% after 216 h because of the poor water-solubility. In contrast, the cumulative release rate of the BNPs reached 92.5% after 216 h due to the decrease of the particle size and the increase of specific surface area.

As far as we know, drug release performance is related to drug solubility. The particle size of the drug prepared by precipitation method was large and uneven, which did not improve the water solubility of the drug, so the release of the drug was relatively slow [38,39,40]. After the nanoparticles were prepared using the template method, the particle size of the drug became smaller, resulting in the increase of specific surface area and enhancement of the solubility, so the drug release was faster and more durable [41,42]. These results indicated that BNPs showed good release properties, raising the dissolution rate of drugs and increasing the utilization efficiency of pesticides.

### 2.4. Stability

The mean particle size and PDI were measured to assess the storage stability of the drug nanoparticles. In Figure 4, the mean particle sizes of BNPs increased to 110 nm and the PDI maintained below 0.3 after storage at 0 °C for 7 days, which means that particles of similar size are more concentrated. The mean particle sizes of BNPs increased to 125 nm and the PDI maintained about 0.3 after storage at 54 °C for 14 days. The particle size of DLS is consistent with that of SEM. As shown in Figure 5, the nanoparticles at different storage temperature exhibited almost spherical morphology and relatively monodisperse distribution.

As mentioned in the literature, the stability of drug particles is related to the uniformity of drug particle size. Small particles tend to grow to large particles, which lead to agglomeration of particles. The higher the disorder degree of particles, the more obvious the aggregation degree of particles. The BNPs has the better uniformity in size, which reduces the aggregation between particles to maintain the stability of particles [43,44,45,46]. In addition, there is no external energy in the whole preparation process, and the surface energy of particles is low, which reduces the aggregation between particles [47]. These results indicated that BNPs showed good storage stability at 0 °C for 7 days and at 54 °C for 14 days.

### 2.5. Promotion and Application

To demonstrate the universality of the current method for pesticides. BNPs with different sizes were prepared using AAO templates with pore size of 20 nm and 200 nm. As shown in Figure 6a, the statistical mean particle size and PDI of nanoparticles using 20 nm AAO template were 22 nm and 0.182, respectively. The mean particle size and PDI of nanoparticles using 200 nm AAO template were 196 nm and 0.201, respectively. The SEM image showed that the particles have spherical shape, uniform size and uniform dispersion.

These results indicated that the particle size of BNPs was limited by the pore size of the template and would not exceed the pore size. The channels in the template are not connected with each other, which prevents the adhesion and growth of particles, so the dispersion and uniformity of particles are very good. In conclusion, the template pore dimension would restrict continuous growth of the nanoparticles, and then determines the size of nanoparticles, which used to explore the properties of drug particles with different sizes.

As depicted in Figure 7, the nanodrugs can be released not only in 0.1 mol/L NaOH solution but also in 1 mol/L diluted hydrochloric acid solution by dissolution of AAO.

The drugs can be collected until the template was completely dissolved in the above alkaline solution for 2 h. In addition, the template can be completely dissolved in 10 h under the above acidic solution. Moreover, it was observed that nanoparticles can be released from AAO template by ultrasonication without dissolving the AAO template. The DLS results showed the particle size was 105 nm in acid solution, and maintained 104 nm in ultrasonic condition.

As known to us, the dissolution of the template did not affect the performance of the drug particles, nor did the ultrasonication. The SEM images showed that these nanoparticles released from the AAO template presented almost spherical and exhibit good dispersion. The AAO templates were completely dissolved in the above NaOH solution and hydrochloric acid solution in supporting Information Appendix A, which avoided the interference of the template to the DLS and SEM results. These results demonstrated that the present method was versatile for hydrophobic drugs and can be easily applied for preparing nanoparticles even if the drugs are sensitive to diluted acid or base.

To demonstrate the versatility of the AAO template-assisted method, we further fabricated nanoparticles of another hydrophobic drugs (avermectin and pyraclostrobin) using the present methods. Avermectin was the representative of insecticide and pyrazoxystrobin was the representative of fungicide. Considering the sensitivity of drugs to acid and base, the ultrasonic method was used to obtain nanoparticles in Figure 8. The DLS results showed the particle size of the avermectin was 108 nm, and the particle size of pyraclostrobin 102 nm.

As known to us, the size of nanoparticles is not affected by different hydrophobic pesticides, which is related to the pore size of the template. The SEM images showed that the nanoparticles were spherical and had good dispersion. In conclusion, the nanoparticles containing multiple drugs can be prepared, which demonstrated that the strategy was versatile and convenient for hydrophobic drugs.

## 3. Materials and Methods

### 3.1. Materials

Buprofezin (97%) was purchased from Hubei Jiufenglong Chemical Co., Ltd (Beiing, China). Anodized aluminum oxide (AAO) membranes were purchased from Beijing Zhongjingkeyi Technology Co., Ltd (Beiing, China). Methanol, Tetrahydrofuran, acetone, hydrochloric acid (HCl) and sodium hydroxide (NaOH) were purchased from Sinopharm Chemical Reagent Co., Ltd (Beiing, China). Milli-Q water (15.0 MΩ cm^−1^, total organic carbon ≤ 4 ppb) was used in all analytical experiments.

### 3.2. Methods

#### 3.2.1. Selection of Organic Solvent

The AAO templates with 100 nm pore size were immersed in methanol, tetrahydrofuran and acetone for several minutes, respectively. The pore size of the template immersed in tetrahydrofuran and acetone increased relative to the theoretical value, which was not suitable for the preparation of nanoparticles. The pore size of the template in methanol did not change, so methanol as a stable solvent was selected for the preparation of buprofezin nanoparticles.

#### 3.2.2. Preparation of Buprofezin Nanoparticles (BNPs) by AAO Method

The BNPs were prepared using AAO template with 100 nm pore size by the following steps. The buprfezin was dissolved in methanol. The AAO templates were first washed sequentially with methanol followed by immersion in buprfezin solutions for 5 min. Then, the soaked templates were still dried at room temperature to remove the organic solvent. During the whole soaking process, the drug was dispersed in the pores of the membrane. The whole process was repeated for several times. The dried templates were dissolved in NaOH solution to dissolve template materials. The discrete nanoparticles were collected by multiple centrifugations and redispersion. The extracted pure nanodrug was finally freeze-dried.

#### 3.2.3. Analysis of BNPs Collection Methods

The BNPs with 100 nm size was collected using 0.1 mol/L NaOH solution to dissolve template materials by multiple centrifugations and redispersion. In order to demonstrate the universality of the current method for drugs. We also explored whether drugs can be released by dissolution of AAO using diluted hydrochloric acid solution (1 mol/L). In addition, we also demonstrated whether the nanoparticles can also be obtained by ultrasonication of the drug-loaded AAO template in water without dissolving the AAO template.

#### 3.2.4. Optimization of Preparation Process of BNPs

Optimization of the preparation process is very important for the size and distribution of drug particles. In this study, particle size and polydispersity index (PDI) were used as the evaluation index to optimize the drug concentration and reaction times. The effect of drug concentration on particle size was investigated in the concentration gradient range of 5, 10, 20 and 40 mg/mL. The effect of soaking and drying cycles on the particle size in drug solutions with a concentration of 20 mg/mL was analyzed by setting 2, 5 and 10 cycles.

#### 3.2.5. Characterization of BNPs

The particle sizes and morphologies of the samples were characterized using SEM (JSM-7401F, JEOL Ltd., Tokyo, Japan). Of the sample solution, 3.5 uL was dripped onto the silicon slice, then dried and sprayed with gold for 40 s. The SEM images were captured at 3 kV voltage and 10 mA current. The morphology of the nanosuspension was characterized by TEM (HT7700, Hitachi Ltd., Tokyo, Japan) with 80 kV accelerating voltage. Two microliters of diluted solution were dripped onto a carbon-coated copper grid and were dried at room temperature for TEM measurement. The hydrated mean particle sizes of the BNPs were examined using dynamic light scattering (DLS, Zetasizer Nano ZS90, Malvern Instruments Ltd., Malvern, UK). The polydispersity index (PDI) was used to characterize particle size distribution. The PDI value less than 0.3 indicated good dispersion. The measurement was carried out in triplicate for each sample.

#### 3.2.6. In Vitro Drug-Release

The drug-release behaviors of BNPs were investigated by ultraviolet visible spectrophotometer (UV, TU1901, Shimadzu, Tokyo, Japan). Five microliters BNPs and free buprofezin were suspended in PBS solution (Ph 7.4, 5 mL) and the solution was transferred to dialysis bags (2000 MWCO), respectively. Finally, the treated dialysis bags were placed in a brown bottle with PBS solution (95 mL) for 216 h. Two milliliters of solution was taken to measure the absorbance by UV spectrophotometer at 246 nm at a specific time. During the dialysis, the solution volume was maintained constant by supply 2 mL of buffer after each sampling. The cumulative release of buprofezin in the solution at different time was calculated according to the standard curve. The assay was performed three times for each sample.

#### 3.2.7. Stability

The BNPs with 100 nm size was selected as an example for stability test. The freeze-dried solid samples were stored at different temperatures (0 °C for 7 days and at 54 °C for 14 days) to explore physicochemical stability. The sample was taken out and diluted 0.5% (*w*/*w*) to determine the hydrated particle size and morphology at a specific time. The stability was evaluated by observing the changes of particle size and PDI during the whole storage process.

#### 3.2.8. Statistical Analysis

The statistical data was presented as mean ± standard deviation (SD). Least significant difference (LSD) was used to analyze data. A probability (p) of less than 0.05 means significant differences.

## 4. Conclusions

In summary, we successfully developed a novel, versatile and controllable approach for solving the problem of dissolution and dispersion of hydrophobic pesticides. The AAO template-assisted method produced nanoparticles of different sizes with excellent monodispersion and uniform morphology, which had better reproducibility and homogeneity comparing with the reprecipitation method. In addition, the pore size of the template restricts the growth of the nanoparticles and then controls the size of the drug particles. The BNPs with the mean size of 100 nm using AAO template-assisted method exhibited better release performance due to the larger specific surface area and faster dissolution rate. The good dispersion and homogeneity of nanoparticles play an important role in maintaining the stability of particles.

In addition, the pore size of the template restricts the continuous growth of the nanoparticles and then controls the size of the drug particles. Furthermore, the nanoparticles could release from the AAO template by dissolving template or ultrasonication even if the drugs are sensitive to diluted acid or base. Moreover, the AAO template method can be applied to other hydrophobic pesticides and extended for fabricating nanoparticles with different functional agents, and used to prepare the nanoparticles containing multiple drugs. In conclusion, the novel AAO template method was not only a versatile and convenient method for the preparing hydrophobic nanodrugs, but also laid a foundation for exploring properties of drug particles with different sizes.

## Figures and Tables

**Figure 1 ijms-22-08348-f001:**
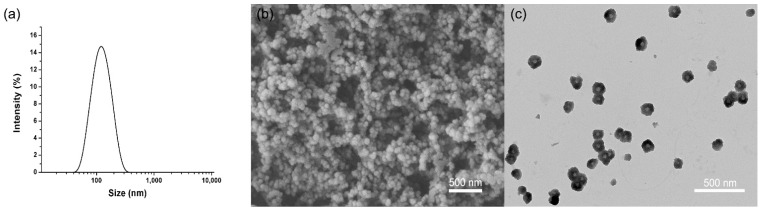
Characterization of particle size and morphology of the buprofezin nanoparticles. (**a**) DLS image of BNPs, (**b**) SEM image of BNPs, (**c**) TEM image of BNPs.

**Figure 2 ijms-22-08348-f002:**
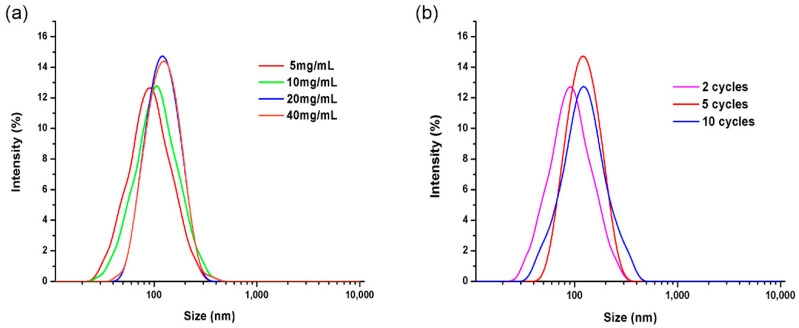
BNPs fabricated with AAO templates with 100 nm pore sizes in different drug concentration (5, 10, 20, 40 mg/mL) with different cycles (2, 5, 10 cycles). (**a**) DLS image of BNPs in different drug concentration (5, 10, 20, 40 mg/mL), (**b**) DLS image of BNPs with different cycles (2, 5, 10 cycles).

**Figure 3 ijms-22-08348-f003:**
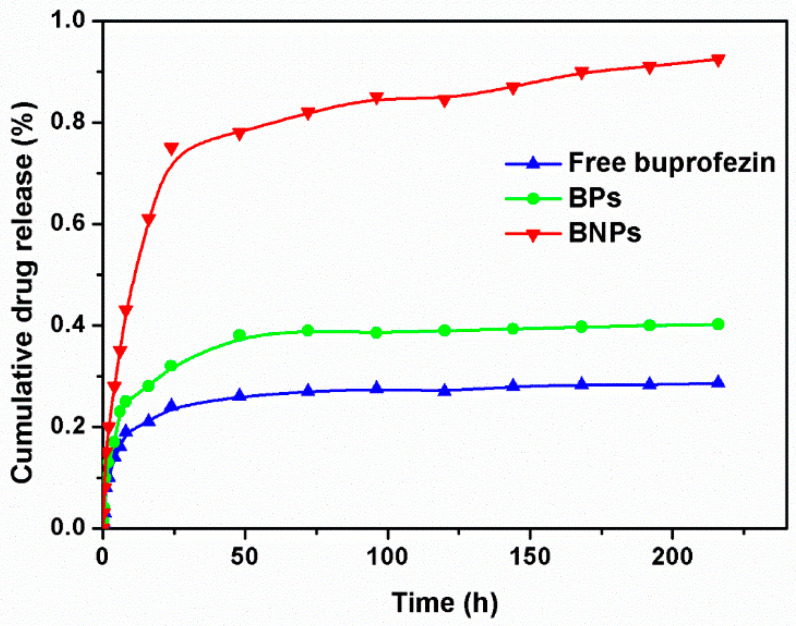
Cumulative drug release from free buprofezin, BPs and BNPs in PBS medium.

**Figure 4 ijms-22-08348-f004:**
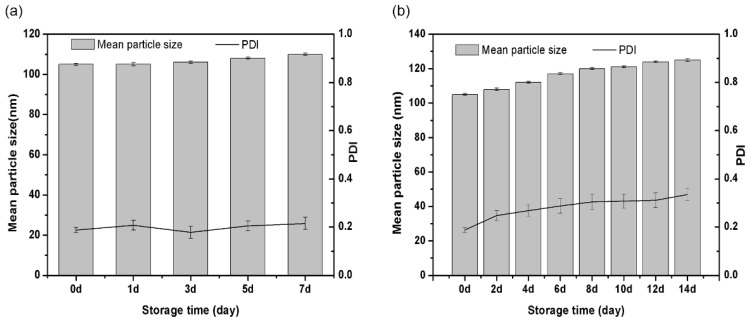
The mean particle size and PDI of the BNPs at different storage temperature. (**a**) BNPs at 0 °C for 7 days, (**b**) BNPs at 54 °C for 14 days.

**Figure 5 ijms-22-08348-f005:**
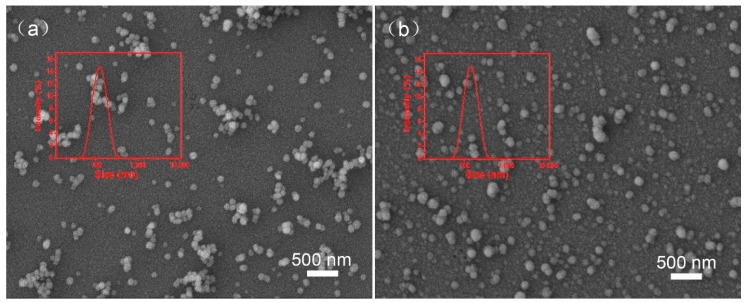
The DLS and SEM image of the BNPs at different storage temperature. (**a**) BNPs at 0 °C for 7 days, (**b**) BNPs at 54 °C for 14 days.

**Figure 6 ijms-22-08348-f006:**
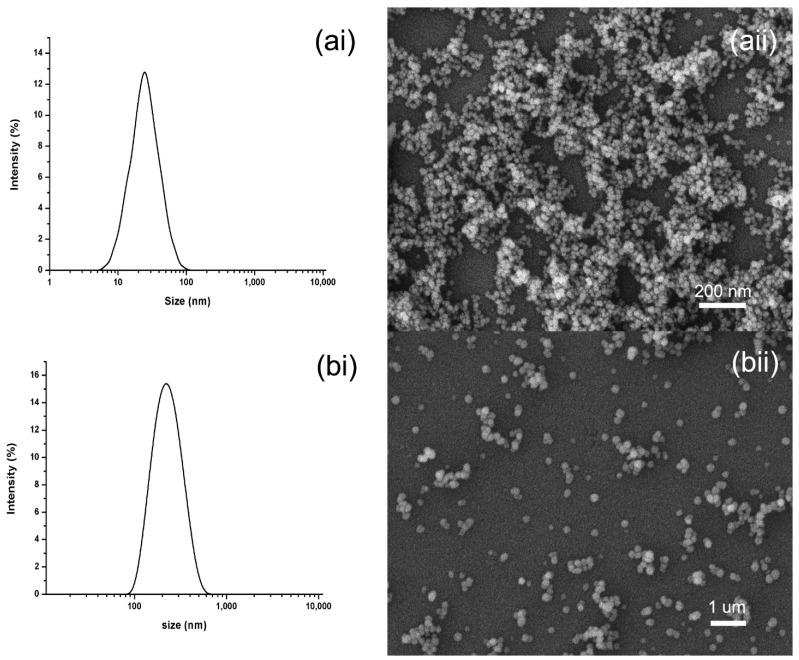
The DLS and SEM characterization of BNPs using AAO templates with pore size of 20 nm and 200 nm. (**ai**) DLS image of BNPs using AAO templates with pore size of 20 nm, (**aii**) SEM image of BNPs using AAO templates with pore size of 20 nm, (**bi**) DLS image of BNPs using AAO templates with pore size of 200 nm, (**bii**) SEM image of BNPs using AAO templates with pore size of 200 nm.

**Figure 7 ijms-22-08348-f007:**
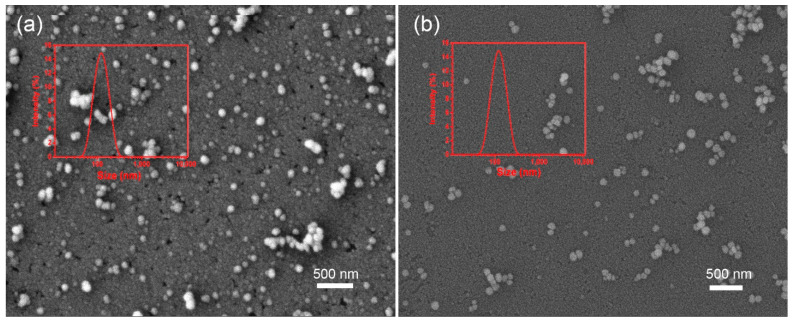
The DLS and SEM characterization of BNPs released from dilute acid and ultrasonic system. (**a**) BNPs released from dilute acid, (**b**) BNPs released from ultrasonic systems.

**Figure 8 ijms-22-08348-f008:**
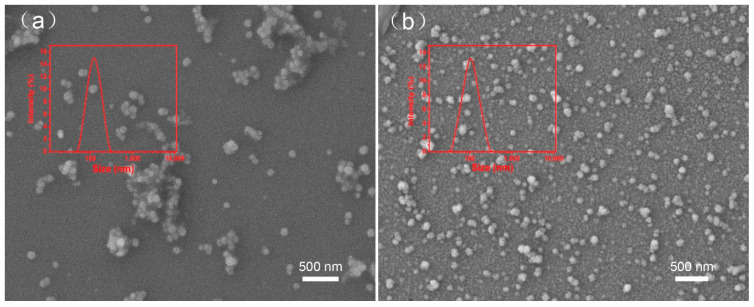
The DLS and SEM characterization of different drug nanoparticles. (**a**) Avermectin, (**b**) pyraclostrobin.

## Data Availability

Not applicable.

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
