# Peer review of "Preparation and Size Control of Efficient and Safe Nanopesticides by Anodic Aluminum Oxide Templates-Assisted Method"

_ijms, 2021, doi:10.3390/ijms22158348_

Round 1

Reviewer 1 Report

The manuscript entitled “Preparation and Size Control of Efficient and Safe Nanopesticides by Anodic Aluminum Oxide Templates-assisted Method” deals with a novel strategy for preparing buprofezin nanoparticles with tunable size via an anodic aluminum oxide template-assisted method. The manuscript is written well overall, and the research is conducted in a scientifically sound manner. Still, I have some serious concerns.

Line 27: “However, the most pesticides are insoluble” – Not true.

Line 62: Is that a nanoparticle?

Section 2.2. It lacks a discussion of why the results are like that.

In general, the results are presented without discussion. Also, some toxicity measurements of the particles should be provided.

Author Response

We have carefully revised the manuscript according to the comments and suggestions. All the amendments have been marked with track changes in the marked-up copy of the revised manuscript. Detailed responses to all the comments are listed below.

Response to Review

Q1: Line 27: “However, the most pesticides are insoluble” – Not true.Re: We thank Reviewer for the helpful comment. The low effective utilization rate of pesticides is a serious problem. These revisions can be seen in line 27-28. 

Q2: Line 62: Is that a nanoparticle?Re: We thank Reviewer for the helpful comment. Buprofezin nanoparticles (BNPs) was used for the first time in line 69. These revisions can be seen in line 69.

Q3: Section 2.2. It lacks a discussion of why the results are like that.

Re: We thank Reviewer for the helpful comment. The particle size of the BNPs with 20 mg/ml was larger than that of the BNPs with 5 and 10 mg/ml. The active components entering into the pore size of the template in unit time increased with the increase of drug concentration,thus enlarging the particle size of the prepared drug particles. The particle size of the particles varied little at the concentration of 20 and 40 mg/ml. In a certain concentration range, the size of drug particles become larger with the increase of the concentration of solution, but it was always controlled in the range of template pore size. These revisions can be seen in line 108-114.

It was also observed that the particle size increased with the number of cycles (2, 5, 10 cycles). The particle size changed little at the number of 5 and 10 cycles. The drug particles will grow up with the increase of the number of cycles, but the size of the particles is always affected by the pore size of the template. These revisions can be seen in line 115-118.

Q4: In general, the results are presented without discussion. Also, some toxicity measurements of the particles should be provided.

Re: We thank Reviewer for the helpful comment. We added discussion after stating the results, and gave a reasonable explanation for the causes of the results. In this study, the preparation process of buprofezin nanoparticles was studied to explore the universality of AAO template method in the preparation of insoluble pesticide nanoparticles. The toxicity of the nanoparticles has been verified in relevant research. The results were as follow: The detection of drug nanoparticles residue is lower than the national limit standard, and the quality of agricultural products is safe (Nanoscale. 2018, 10, 1798-1806). 2,4-D sodium salt (2,4-dichlorophenoxyacetic acid sodium salt) nanodrug delivery system is safe for non-target crop wheat, and has good biological activity for broad-leaved crop cucumber (Journal of Agricultural and Food Chemistry. 2018, 66, 6594-6603). According to the exposure assessment method, the nano pesticide drug delivery system does not increase the exposure risk to the operator (Science of the Total Environment. 2018, 624,1195-1201).

Reviewer 2 Report

The manuscript by Wang et al. “Preparation and Size Control of Efficient and Safe Nanopesticides by Anodic Aluminum Oxide Templates-assisted Method” is well presented. There are minor concerns to be addressed before its publication as follows:

Comments.

  1. The novelty and objectives of this study may be stated clearly at the end of the Introduction section.
  2.  The introduction may be improved with quantitative data of potential application of Nanopesticides; and advantages & disadvantages of anodic aluminum oxide (AAO) template-assisted method with few literature details; and Is this can be used for large-scale/pilot synthesis?
  3. Quality of Figures may be improved like resolutions, font sizes, line widths.
  4. A comparative Table may be provided to highlight the significance of this study with literature reports based on related studies.
  5. Discussion may be polished more with recent citations.

Author Response

We have carefully revised the manuscript according to the comments and suggestions. All the amendments have been marked with track changes in the marked-up copy of the revised manuscript. Detailed responses to all the comments are listed below.

Response to Review

Q1: The novelty and objectives of this study may be stated clearly at the end of the Introduction section. Re: We thank Reviewer for the helpful comment. In this study, the buprofezin nanoparticles (BNPs) with different sizes were developed via AAO template-assisted process. The particles with similar surface characteristics and extremely narrow size distribution were obtained using AAO templates with different pore sizes, which could also exclude the impact of shape and surface charge compared with the traditional method. The effect of drug concentration and number cycles on the size of particle was investigated. At the same time, the stability and release properties of nanoparticles were evaluated to characterize the effects of particle size and dispersion on their physicochemical properties. In conclusion, this research not only established an innovative and simple method for the potential application of nanopesticides with controllable particle size, excellent monodispersity and uniform morphology in plant protection, but also laid a foundation for the study of the physicochemical properties of drug particles with different sizes. These revisions can be seen in line 69-81.

Q2: The introduction may be improved with quantitative data of potential application of Nanopesticides; and advantages & disadvantages of anodic aluminum oxide (AAO) template-assisted method with few literature details; and Is this can be used for large-scale/pilot synthesis?

Re: We thank Reviewer for the helpful comment. Some examples of nano pesticide are added. Functionalized abamectin poly (lactic acid) nanoparticles were prepared by the ultrasonic emulsification to explore the adhesion of nanoparticles to the leaf surface of crops. Antagonistic effect of azoxystrobin poly (lactic acid) microspheres with controllable particle size was studied. These studies showed a large influence of the preparation conditions on the drug particles [16, 17]. These revisions can be seen in line 39-43. Pyrimethanil-loaded mesoporous silica nanoparticles was synthesis to explore the Distribution and Dissipation in Cucumber Plants. Abamectin using porous silica nanoparticles as carriers were constructed to evaluate controlled-release performance [23, 24]. These revisions can be seen in line 44-49.

As known to us, anodized aluminum oxide (AAO) templates have been widely used for forming nanotubes and nanowires. The tiny holes on the surface of AAO are arranged in an orderly manner. In addition, the pore channels of the AAO template were parallel to each other, and the pore size uniformity of the template was good. Based on the above principle, the AAO template has been applied for forming nanoparticles by making full use of the characteristics of AAO template. However, AAO templates have never been used for preparing pesticide nanoparticles. In this work, we demonstrated for the first time that the AAO template method can in fact be extended for preparing uniform pesticide nanoparticles comparing to the conventional reprecipitation approach [25-28]. These revisions can be seen in line 52-60.

Q3: Quality of Figures may be improved like resolutions, font sizes, line widths.

Re: We thank Reviewer for the helpful comment. We have revised it according to the standard of the magazine. We have improved the resolution for Figure 5, Figure 7 and Figure 8.

Q4: A comparative Table may be provided to highlight the significance of this study with literature reports based on related studies.

Re: We thank Reviewer for the helpful comment. The DLS and SEM image of the buprofezin particles (BPs) using reprecipitation method in Figure 2S, which shows the advantages of the template method compared with the traditional method. The comparisions between AAO template method and reprecipitation methods were as follow:

Table1. The comparison between AAO template method and reprecipitation methods

Characterization

AAO template method

Reprecipitation method

Particle size

Controllable

Uncontrollable

Uniformity

Monodispersity

Polydispersity

Dispersibility

Uniform dispersion

Uneven dispersion

Applicability for pesticide

High

General

Q5: Discussion may be polished more with recent citations.

Re: We thank Reviewer for the helpful comment. We added the recent citations including reference 41 and 42 to explains the reason why the cumulative release rate of the BNPs is relatively high. These revisions can be seen in line 136. In addition, the main reason of the BNPs with good stability may be that the particle size of the nanoparticles has the better uniformity, which reduces the aggregation between particles [43-46]. In addition, there is no external energy in the whole preparation process, and the surface energy of particles is low [47]. These revisions can be seen in line 149-151.

Round 2

Reviewer 1 Report

Except for part 2.2, The manuscript is still completely lacking the discussion. In my opinion, it must be improved. 

Author Response

 We have carefully revised the manuscript according to the comments and suggestions. All the amendments have been marked with track changes in the marked-up copy of the revised manuscript. Detailed responses to all the comments are listed below.

Response to Review

Q: Except for part 2.2, The manuscript is still completely lacking the discussion. In my opinion, it must be improved.Re: We thank Reviewer for the helpful comment. We added the discussion after the results. The discussions were as follows:

For part 2.1.

These results indicated that BNPs exhibited almost spherical morphology and comparatively monodisperse distribution. As known to us, the size of each aperture of the template is consistent, and the template is arranged in order, so the prepared drug particles are uniform in size. During the whole preparation process, drug molecules were gradually deposited in the pore of AAO template, and the particle size was limited by the pore size finally [25, 27]. Therefore, the template with different pore size can be used to prepare nanoparticles with different sizes, which laid a foundation for the study of the physicochemical properties of drug particles with different sizes. These revisions can be seen in line 88-95. 

For part 2.3.

As far as we know, drug release performance is related to drug solubility. The particle size of the drug prepared by precipitation method was large and uneven, which did not improve the water solubility of the drug, so the release of the drug was relatively slow [38-40]. After the nanoparticles were prepared using the template method, the particle size of the drug became smaller, resulting in the increase of specific surface area and enhancement of the solubility, so the drug release was faster and more durable [41, 42]. These results indicated that BNPs showed good release properties, raising the dissolution rate of drugs and increasing the utilization efficiency of pesticides. These revisions can be seen in line 144-151.

For part 2.4.

As mentioned in the literature, the stability of drug particles is related to the uniformity of drug particle size. Small particles tend to grow to large particles, which lead to agglomeration of particles. The higher the disorder degree of particles, the more obvious the aggregation degree of particles. The BNPs has the better uniformity in size, which reduces the aggregation between particles to maintain the stability of particles [43-46]. In addition, there is no external energy in the whole preparation process, and the surface energy of particles is low, which reduces the aggregation between particles [47]. These results indicated that BNPs showed good storage stability at 0℃ for 7 days and at 54℃for 14 days. These revisions can be seen in line 166-174.

For part 2.5.

These results indicated that the particle size of BNPs was limited by the pore size of the template and would not exceed the pore size. The channels in the template are not connected with each other, which prevents the adhesion and growth of particles, so the dispersion and uniformity of particles are very good. In conclusion, the template pore dimension would restrict continuous growth of the nanoparticles, and then determines the size of nanoparticles, which used to explore the properties of drug particles with different sizes. These revisions can be seen in line 191-197.

The drugs can be collected until the template was completely dissolved in the above alkaline solution for 2 hours. In addition, the template can be completely dissolved in 10 h under the above acidic solution. Moreover, it was observed that nanoparticles can be released from AAO template by ultrasonication without dissolving the AAO template. The DLS results showed the particle size was 105 nm in acid solution, and maintained 104 nm in ultrasonic condition. These revisions can be seen in line 204-209.

As known to us, the dissolution of the template did not affect the performance of the drug particles, nor did the ultrasonication. The SEM images showed that these nanoparticles released from the AAO template presented almost spherical and exhibit good dispersion. The AAO templates were completely dissolved in the above NaOH solution and hydrochloric acid solution in supporting Information Figure S3, which avoided the interference of the template to the DLS and SEM results. These results demonstrated that the present method was versatile for hydrophobic drugs and can be easily applied for preparing nanoparticles even if the drugs are sensitive to diluted acid or base. These revisions can be seen in line 210-217.

The DLS results showed the particle size of the avermectin was 108 nm, and the particle size of pyraclostrobin 102 nm. As known to us, the size of nanoparticles is not affected by different hydrophobic pesticides, which is related to the pore size of the template. The SEM images showed that the nanoparticles were spherical and had good dispersion. In conclusion, the nanoparticles containing multiple drugs can be prepared, which demonstrated that the strategy was versatile and convenient for hydrophobic drugs. These revisions can be seen in line 237-243.

For  conclusion.

In summary, we successfully developed a novel, versatile and controllable approach for solving the problem of dissolution and dispersion of hydrophobic pesticides. The AAO template-assisted method produced nanoparticles of different sizes with excellent monodispersion and uniform morphology, which had better reproducibility and homogeneity comparing with the reprecipitation method. We could tailor the morphology and structure of nanoparticles by regulating the preparation conditions including the drug concentration and the cycle numbers, thus control drug release and maintain storage stability.  The BNPs with the mean size of 100 nm using AAO template-assisted method exhibited better release performance due to the larger specific surface area and faster dissolution rate. The good dispersion and homogeneity of nanoparticles play an important role in maintaining the stability of particles.

In addition, the pore size of the template restricts the continuous growth of the nanoparticles and then controls the size of the drug particles. Furthermore, the nanoparticles could release from the AAO template by dissolving template or ultrasonication even if the drugs are sensitive to diluted acid or base. Moreover, the AAO template method can be applied to other hydrophobic pesticides and extended for fabricating nanoparticles with different functional agents, and used to prepare the nanoparticles containing multiple drugs. In conclusion, the novel AAO template method was not only a versatile and convenient method for the preparing hydrophobic nanodrugs, but also laid a foundation for exploring properties of drug particles with different sizes. These revisions can be seen in line 301-320.

Round 3

Reviewer 1 Report

The authors added the discussion. I recommend the manuscript for publication in the present form.